# A Single Amino Acid Substitution in MIL1 Leads to Activation of Programmed Cell Death and Defense Responses in Rice

**DOI:** 10.3390/ijms23168853

**Published:** 2022-08-09

**Authors:** Bowen Yan, Haoyu Zheng, Yuwei Sang, Yan Wang, Jian Sun, Fengcheng Li, Jiayu Wang, Xiaoxue Wang

**Affiliations:** 1Rice Research Institute, Shenyang Agricultural University, Shenyang 110866, China; 2College of Plant Protection, Shenyang Agricultural University, Shenyang 110866, China

**Keywords:** lesion mimic mutants, *mil1-1*, programmed cell death, disease resistance, rice

## Abstract

Lesion mimic mutants are an ideal model system for elucidating the molecular mechanisms of programmed cell death and defense responses in rice. In this study, we identified a lesion mimic mutant termed *miner infection like 1-1* (*mil1-1*). The *mil1-1* exhibited lesions on the leaves during development, and the chloroplasts of *mil1-1* leaves were disrupted. Reactive oxygen species were found to accumulate in *mil1-1* leaves. Cell death and DNA fragmentation were observed in *mil1-1* leaves, indicating that the cells in the spots of *mil1-1* leaves experienced programmed cell death. Most agronomic traits decreased in *mil1-1*, suggesting that the growth retardation in *mil1-1* caused reduced per-plant grain yield. However, the mutation of *MIL1* activated the expression of pathogen response genes and enhanced resistance to bacterial blight. The *MIL1* gene was cloned using the positional cloning approach. A missense mutation 751 bp downstream of ATG was found in *mil1-1*. The defects of *mil1-1* were able to be rescued by delivering a wild-type *MIL1* gene into *mil1-1*. *MIL1* encoded hydroperoxide lyase 3 (OsHPL3), and the expression of *OsHPL3* was induced via hormone and abiotic stresses. Our findings provide insights into the roles of MIL1 in regulating programmed cell death, development, yield, and defense responses in rice.

## 1. Introduction

Rice is one of the primary food crops globally, feeding more than half of the population of the world. The grain yield of rice is also closely associated with global food security. Disease is one of the most important factors limiting the yield and quality enhancement of rice. In response to pathogen attacks, a hypersensitive response (HR), which induces infected cell death, prevents the spread of pathogens to adjacent cells. The over-accumulation of reactive oxygen species (ROS), including superoxide radical (O^2−^) and hydrogen peroxide (H_2_O_2_), has been confirmed to be closely associated with lesion formation in several lesion mimic mutants [1,2].

In plants, ROS are generated during basal metabolic processes and play important roles in the development and responses to various abiotic and biotic stresses [3]. Most apoplastic ROS are produced via respiratory burst oxidase homologs D (RbohD). Superoxide anion radicals (O^2−^), hydroxyl radicals (OH^−^), and hydrogen peroxide (H_2_O_2_) are the major ROS molecules leading to oxidative damage in plants [4]. Under normal physiologic conditions, cells control ROS levels by balancing the generation of ROS with their elimination using the scavenging system. ROS are regarded as signal molecules that participate in programmed cell death (PCD). Accumulated ROS can directly damage normal cellular structures or act as signal molecules to induce HR, resulting in cell death and necrotic lesions in leaves [5]. When a pathogen invades a plant, HR is generated, leading to PCD and thereby suppressing pathogen infection.

Lesion mimic mutants display HR-like spontaneous lesions in the absence of pathogen attacks, abiotic stresses, and mechanical damage, which are useful to uncover the regulatory mechanisms of defense responses and PCD in rice [6]. Recently, several of the genes involved in defense responses to pathogen infection and PCD have been identified by using lesion mimic mutants in rice [6]. Lesion mimic mutations in rice affect spot-leaf formation, PCD, and resistance to disease via different molecular mechanisms.

Knockout of *Spotted Leaf 3* (*SPL3*) and *Lesion Mimic Mutant 24* (*LMM24*) genes, which encode Mitogen-Activated Protein Kinase Kinase Kinase 1 (OsMAPKKK1) and Receptor-Like Cytoplasmic Kinase 109 (RLCK109), respectively, develop spontaneous lesions on the leaf blades and enhance resistance to bacterial blight [7,8,9,10,11]. Mutations in the genes encoding AAA-type ATPase, such as *SPL4* and *Lesion Mimic Resembling* (*LMR*), exhibit spontaneous lesions and affect leaf senescence, immune response, and PCD in rice [12,13,14]. Mutations of genes involved in protein synthesis, such as *SPL33* and *Lesion Mimic Leaf 1* (*LML1*), encoding translation Elongation Factor 1 alpha (eEF1A) and eukaryotic Release Factor 1 (eRF1), respectively, display spontaneous lesions in leaves [15,16]. Mutations of genes encoding proteins in the poly-ubiquitination mediated 26S proteasome system, such as SPL11 (an E3 ubiquitin ligase), OsCUL3a (a component in SKP1-CUL1-F-box protein E3 ligase complex), and SPL35 (a coupling of ubiquitin binding to endoplasmic reticulum degradation domain protein), present lesion-like leaves [17,18,19]. Mutants of several genes, such as *Lesion Like Mutant 1* (*LLM1*), *SPL29*, and *SPL30*, which encode coproporphyrinogen III oxidase (CPOX), uridine diphosphate-N-acetylglucosamine pyrophosphorylase (UAP1), and P450 monooxygenase, respectively, also exhibit different lesion mimic phenotypes in rice [20,21,22,23].

Defense responses of plants against pathogen attacks are complex processes. More questions remain to be uncovered. Cloning and characterization of genes involved in defense responses with lesion mimic mutants can further dissect the mechanisms of ROS generation, PCD, and responses to pathogen attacks in plants. In this study, we obtained a lesion mimic mutant, termed *miner infection like 1-1* (*mil1-1*), using an ethyl methane sulfonate (EMS) mutagenesis approach in rice. The *mil1-1* mutant exhibits spontaneous lesions in leaf blades from the early seedling stage to the grain filling stage. The emergence of these spots in *mil1-1* is light-intensity dependent. ROS are abnormally accumulated causing PCD in *mil1-1* leaves. Additionally, the *mil1-1* mutant displays enhanced resistance to bacterial blight, indicating that MIL1 negatively regulates resistance to bacterial blight disease in rice. The *MIL1* gene was isolated through the map-based positional cloning approach. A single-base substitution, G to A, in 751 bp downstream of ATG at the Os02g0110200 locus of *mil1-1*, changing the codon from GGG (glycine, Gly) to AGG (arginine, Arg), confers the phenotype of *mil1-1*. The *MIL1* gene is known to encode Hydroperoxide Lyase 3 (OsHPL3) in rice.

## 2. Results

### 2.1. The Mil1-1 Mutation Confers Spotted Leaf Phenotype in Rice

The spotted leaf mutant of rice, *mil1-1*, was obtained by EMS mutagenesis. From the early seedling stage to the mature stage, *mil1-1* displayed a number of spontaneous lesions in the leaves (Figure 1A,B).

Because chloroplasts are fragile. Color change and lesion formation in leaves will destroy the integrality of chloroplasts, especially the structure of thylakoid lamellae [11]. To determine whether *mil1-1* mutation disrupts the leaf microstructure, the ultrastructure of chloroplasts in the mesophyll cells was observed using transmission electron microscopy (TEM). The chloroplasts were found to be well-developed, and the thylakoid lamellae were regular and compact in the WT sample; however, the thylakoid lamellae were loosely arranged in *mil1-1*, indicating that the chloroplasts in *mil1-1* were destroyed by the formation of lesions in the leaf blade (Figure 1C).

Lesion formation of lesion mimic mutants is usually light-intensity dependent [24]. To explore the effects of light intensity on *mil1-1* spot production, 70 percent of shade treatments were applied for two weeks at the tillering stage in the paddy field. The results showed that visible spots continued to develop on leaves of *mil1-1* mutant under normal conditions (no shade); however, the leaves of *mil1-1* mutant no longer developed spots after shade treatment, and the spots on the *mil1-1* leaves before shade treatment did not vanish (Figure 1D). These findings suggest that light intensity influences the establishment of spots in the leaves of *mil1-1*. In addition, the difference in the photosynthetic rate between WT and *mil1-1* was investigated. The photosynthetic rate of *mil1-1* was found to be slightly lower than that of WT, but the difference was not significant (Figure 1E).

### 2.2. Abnormal ROS Accumulation and PCD Occur in Mil1-1

To understand the molecular mechanisms driving the formation of HR-like lesions in the leaves of *mil1-1*, we analyzed the expression of numerous histochemical markers for ROS accumulation and cell death. Excessive buildup of H_2_O_2_ in the tissue may generate a reddish-brown precipitate with 3,3′-diaminobenzidine (DAB). Based on this factor, the leaves from 60-day-old plants of WT and *mil1-1* were selected for DAB staining. The results showed that H_2_O_2_ accumulated in the leaves of *mil1-1* but not in the leaves of WT (Figure 2A). Similarly, purple dirty precipitation formed in *mil**1-1* stained with nitro-blue tetrazolium chloride (NBT), which is an indicator of O^2−^, but not in any WT leaves (Figure 2B). Consistent with the DAB staining, the H_2_O_2_ content in *mil1-1* was significantly higher than that in WT, indicating that H_2_O_2_ accumulated to a higher level in *mil1-1* (Figure 2C).

In addition, the activities of peroxidase (POD) and catalase (CAT) were examined to explore the effects of *mil1-1* mutation on the ROS scavenging system. The results revealed that the activities of POD and CAT were lower in *mil1-1* than in WT, suggesting that the ROS scavenging system was weakened by *mil1-1*, causing the accumulation of ROS (Figure 2D,E).

To confirm whether the lesions in the leaf blade of *mil1-1* indicated cell death, trypan blue staining was conducted. Cells in the spotted region of the *mil1-1* leaves turned dark blue when the leaves were stained with trypan blue, whereas no dark blue spots were observed in WT leaves, indicating irreversible membrane damage or cell death in the spotted leaves of *mil1-1* (Figure 2F). By using a terminal deoxynucleotidyl transferase-mediated dUTP nick-end labeling (TUNEL) method, DNA fragmentation, which is a PCD indicator, was detected. TUNEL signals were observed in *mil1-1* but not in WT plants (Figure 2G). These results indicated that ROS accumulation was closely related to cell death and the lesion mimic phenotype in *mil1-1*.

### 2.3. Mil1-1 Mutation Impacts Pleiotropic Agronomic Traits

To analyze the effects of *mil1-1* mutation on rice yield, agronomic traits were investigated. The plant height of *mil1-1* was significantly shorter than that of the WT (Figure 3A,B), and the number of tillers was reduced in *mil1-1* (Figure 3C). To determine whether *mil1-1* affected biomass production, the dry weight and fresh weight of different organs above ground were measured. Compared to WT, dry matter accumulation was significantly reduced in *mil1-1*, especially the dry weight of the stem and panicle. Similarly, the fresh weight of *mil1-1* was significantly lower than that of WT (Figure 3D). The economic coefficient of *mil1-1* was lower than that of WT (Figure 3E).

In addition, we examined the panicle and grain traits in this study. The pleiotropic traits of *mil1-1*, including panicle length, primary and secondary branch number, total grain number, filled grain number, seed-setting rate, 1000-grain weight, grain width, and grain thickness, were significantly decreased compared to those of WT (Figure 4A–O). The grain yield per plant was significantly reduced via *mil1-1* mutation (Figure 4P). Taken together, these results suggest that the spontaneous lesion mimic phenotypes of *mil1-1* are accompanied by significant decreases in major agronomic traits.

### 2.4. Mutation of MIL1 Enhances Bacterial Blight Resistance

In some spotted leaf mutants of rice, activation of the immune response and enhanced resistance to pathogen infection were previously reported [15]. To explore whether *mil1-1* mutant also exhibits disease resistance phenotypes, we inoculated *mil1-1* mutant, WT, and Li Jiang Hei Gu (LJHG, a variety susceptible to a wide variety of pathogens) with the bacterial blight (*Xanthomonas oryzae* pv. *Oryzae*, *Xoo*) isolate Zhe173, which is virulent against WT. The results showed that *mil1-1* mutation conferred disease-resistant phenotypes against bacterial blight (Figure 5A). The disease incidence rate and the lesion length in *mil**1-1* were higher and longer than those in WT (Figure 5B,C).

Activation of defense response genes was previously observed during the development of lesions in some rice lesion mimic materials [2,23,25,26]. To investigate whether defense response genes are activated in *mil1-1* mutant, the expression of defense response genes, including three pathogen response (PR) genes (*PR1a*, *PR1b*, and *PBZ1/PR10*) and two pathogen-induced peroxidase genes (*POX1/POX22.3* and *PO-C1*), was examined using RT-qPCR before and after lesion formation in *mil1-1*. Before lesion formation, expression of the five defense response genes was lower than that of WT (Figure 5D–H); however, after lesion developed on the leaves of *mil1-1*, expression of the genes was strongly induced by *mil1-1*, with significantly higher results than those in WT (Figure 5I–M). These results indicate that the mutation of *MIL1* activates the expression of defense response genes after lesion formation and enhances disease resistance to bacterial blast.

### 2.5. Mil1-1 Is a Novel Allelic Mutation at OsHPL3 Locus

In order to clone the *mil1-1* mutation, a genetic analysis was performed. The *mil1-1* mutant was crossed to WT. All the first filial generation (F_1_) hybrids showed normal phenotypes (i.e., the *mil1-1* phenotype was not observed in F_1_). In the second generation (F_2_) segregation population, 210 plants exhibited the *mil1-1* phenotype, and 593 plants showed a normal phenotype. The ratio of individuals with a normal phenotype to those with the *mil1-1* phenotype was 2.81:1, χ^2^ = 0.5683 < χ^2^ (*P*_0.05,1_ = 3.84), indicating good agreement with the expected value of 3:1. These results indicate that *mil1-1* is a single recessive mutation (Table 1).

Since japonica and indica rice are subspecies of cultivated rice, molecular polymorphisms between their genomes are more abundant. To isolate the *MIL1* gene, an F_2_ segregation population was created by crossing *mil1-1* (in japonica background) with the indica variety Habataki to facilitate developing molecular markers for map-based cloning of *mil1-1*. Bulked Segregant Analysis (BSA) was used to preliminarily locate the *mil1-1* mutation. In the F_2_ segregation population, 41 individuals with the *mil1-1* mutant phenotype were selected to create a DNA pool. DNA from the F_1_ plants (*mil1-1* × Habataki hybrid) was used as a control. Polymorphic molecular markers evenly distributed on the genome, such as InDel and SSR markers, were used for preliminary mapping. The results showed that *mil1-1* was mapped between G2-1 and B2-3 molecular markers on chromosome 2 (Figure 6A). By using 251 individuals with the *mil1-1* mutant phenotype in the F_2_ population, the *MIL1* locus was fine-mapped to a 243 kb region flanked by the GN2-1 and GN2-2 markers, in which there were 18 open reading frames (ORFs). The ORFs in the region were then sequenced. A single base substitution (G to A) at 751 bp downstream of ATG was identified in ORF3 (Os02g0110200), which changed the codon from GGG to AGG, causing a shift of glycine (Gly) to arginine (Arg) (Figure 6A,B).

To verify whether the mutation of *MIL1* was responsible for the lesion mimic phenotype of *mil1-1*, a complementary test was performed by transforming the *35S:**MIL**1-GFP* plasmid into *mil**1-1*. All of the independent transgenic lines exhibited a rescued phenotype; three of them (C-1, C-2, and C-3 complementary transgenic lines) were chosen for further study. The spontaneous lesion mimic phenotype of *mil1-1* was rescued in the three complementary transgenic lines, confirming that the missense point mutation in Os02g0110200 locus confers the phenotype of *mil1-1* (Figure 6C,D). The Os02g0110200 gene encodes Hydroperoxide Lyase 3 (OsHPL3) protein belonging to the cytochrome P450 superfamily.

Genes regulating tolerance to abiotic stresses are commonly activated by stress conditions. Crosstalk usually exists in tolerance to biotic and abiotic stresses. MIL1 is involved in resistance to bacterial blight, implying that MIL1 may be associated with tolerance to abiotic stresses. To study whether *MIL1* is induced under stress conditions, 14-day-old seedlings were sampled to determine the expression of *MIL1*/*OsHPL3* responses to jasmonic acid (JA), salicylic acid (SA), abscisic acid (ABA), salt, and wounding. The results showed that *MIL1*/*OsHPL3* was obviously induced by the treatments, suggesting that *MIL1* may be related to tolerance to abiotic stresses (Figure 6E).

## 3. Discussion

Almost all of the spontaneous spotted leaf mutants or lesion mimic mutants display similar phenotypes without pathogen attack in rice though the color, size, shape of the spots, and functions of mutations differ one from the other. They play important roles in deciphering the mechanisms of ROS accumulation, PCD formation, leaf senescence, and defense response [6]. In this study, a novel allele of lesion mimic mutant, *mil1-1*, was characterized.

### 3.1. Mutation of MIL1 Increases the Accumulation of ROS Causing Spontaneous Lesions in Leaves

Lesion mimic mutants have been extensively studied to reveal the principles of ROS burst and PCD in rice [6]. Cells in the region of the spontaneous lesions usually undergo PCD triggered by the accumulation of ROS [27,28]. In this study, abnormal accumulation of ROS was observed in *mil1-1* through histochemical staining and a chemical assay (Figure 2A–C). The eruption of ROS destroyed the chloroplasts and leaf cells, causing PCD and lesion formation in the leaves of *mil1-1* (Figure 1A–C and Figure 2F,G).

The ROS are formed in organelles during electron transport reactions in photosynthesis and respiration and also as byproducts of enzymatic reactions in photorespiration and other metabolic processes. Plants produce more ROS under light than under shade or dark conditions [29]. If the ROS scavenging system is disrupted or weakened, ROS will accumulate to damage the cells. This phenomenon is consistent with our observation that the spot formation in *mil1-1* was light-intensity dependent, as no lesions formed in *mil1-1* under shade conditions (Figure 1D). The activities of ROS scavenging enzymes, such as POD and CAT, were reduced in *mil1-1*, resulting in elevated levels of ROS in *mil1-1* leaves (Figure 2D,E).

### 3.2. Mutation of MIL1 Leads to Auto-Activation of Defense Response in Rice

Like other plants, rice is exposed to diverse pathogens. To counter the challenges of pathogenic infection, rice has developed immune systems to recognize and counteract these pathogens to ensure survival [30]. Two layers of immune systems have evolved in plants and rice. Pattern recognition receptors (PRRs) on the cell surface trigger the primary immune response or pattern-triggered immunity (PTI), whereas intracellular nucleotide-binding oligomerization domain-containing-leucine rich repeat (NLR) proteins act as factors that recognize effectors secreted by pathogens and induce effector-triggered immunity (ETI) [31,32]. Defense responses, including HR, ROS burst, Ca^2+^ influx, kinase activation, and transcriptional activation of defense response genes, act against pathogen attacks [33].

In this study, we found that the spontaneous lesions in *mil**1-1* leaves without pathogen infection were HR-like (Figure 2F,G). Moreover, the *mil1-1* mutant exhibited resistance to bacterial blight (Figure 5A–C). The expression of defense response genes was significantly activated in *mil1-1* after lesions were well-developed (Figure 5I–M). These results suggest that the auto-activation of defense responses in *mil1-1*, including higher accumulation of ROS and PCD, are responsible for the resistance of *mil1-1* to disease, which is consistent with previous reports [16,19].

### 3.3. Mil1-1 Retards Development and Reduces Grain Yield of Rice

Plant immune systems are accurately regulated by multiple negative regulators to avoid unnecessary defense activation. The immune regulators maintain an inactive status in the absence of pathogens and become activated only upon pathogenic challenges. However, loss-of-function mutations of the negative regulators or gain-of-function mutations of positive regulators, such as NLRs, can trigger defense activation, resulting in autoimmunity. Auto-activation of the defense system is usually accompanied by yield reduction [34,35].

In this study, pleiotropic agronomic traits were affected by *mil1-1*, and the plant height and biomass production were reduced in *mil1-1* (Figure 3A,B,D). Panicle traits, such as the panicle length, panicle weight, and the number of primary and secondary branches, decreased in *mil1-1* compared to WT (Figure 4A–E). The total grain number, filled grain number, seed setting rate, and 1000-grain weight in *mil1-1* were lower than those in WT, causing a reduced per-plant grain yield of *mil1-1* (Figure 4). The retarded development and reduced grain yield in *mil1-1* may be related to the auto-activation of defense responses because this process consumes energy [36]. Suppressing the continuous activation of resistant mechanisms adopted by plant species against pathogens is thus a wise option to protect plants from yield penalties.

### 3.4. Mutation of MIL1 Is a Novel Missense Point Mutation of OsHPL3

Mutation of the *MIL1* locus was previously reported as *hpl3*-*1* and *cea62* [37,38]. The *hpl3-1* mutant, with a transposon insertion in its coding region, showed a lesion-like phenotype in the leaves of 2-week-old seedlings and onwards [38]. The *cea62* is another mutation at *OsHPL3*, in which tyrosine (Tyr) at 382 amino acid is changed to a stop codon [37]. These mutants lack hydroperoxide lyase activity. In this study, the *mil1-1* mutation was isolated using the map-based cloning approach. One base pair G to A mutation at 751 bp downstream of ATG was found in *MIL1* (Os02g0110200) locus, which resulted in the replacement of Gly at the 295 amino acid with Arg (G295A) (Figure 6). A single amino acid substitution in the *MIL1* alters its functions and confers disease-like phenotypes in *mil1-1* mutant.

In conclusion, a novel allele of *oshpl3* termed *mil1-1* was isolated in this study. The auto-activation of defense responses in *mil1-1*, including the higher accumulation of ROS and PCD, resulted in resistance to disease and tolerance to salt stress. Although lesion mimic mutants of rice have undesirable agronomic traits, the genetic principles of lesion-like phenotypes can be used to obtain high grain yields by deciphering the efficiency of photosynthesis, disease resistance, and environmental stress responses. Disease-like spot mutants are invaluable germplasm resources for disease-resistance breeding in rice.

## 4. Materials and Methods

### 4.1. Plant Materials, Growth Conditions, and Agronomic Trait Evaluation

The japonica rice variety Shen Nong 9816 (SN9816), the Japanese indica variety Habataki, and the *mil1-1* mutant were used as materials in this study. SN9816 was used as the wild type (WT) to produce a population for genetic analysis. Habataki was used to generate the mapping population. The *mil1-1* mutant was obtained by screening the M_2_ population of EMS mutagenesis [39]. In brief, the seeds of WT were soaked with water. The soaked seeds were then treated using 0.6% EMS. After cleaning with water, the seeds were germinated and sowed on the seedbeds to nurse seedlings. At the four-leaf stage, seedlings were transplanted to a paddy field and labeled as the M_1_ population. M_1_ plants were self-pollinated. Seeds from individual M_1_ plants were harvested to generate the M_2_ segregation population for mutant screening.

WT and *mil1-1* mutant plants were grown on a paddy field in Shenyang, China (Longitude: 123.38° E; Latitude: 41.8° N). Rice seeds used in the experiment were sterilized and germinated under humid conditions at 28–30 °C. After germination, the seeds were sown into seedbeds. Seedlings were then transplanted to the paddy field under natural growth conditions during the four-leaf stage. All materials were grown in the paddy field from April to October.

To measure major agronomic traits, 3 replicates were performed in the paddy field. Twenty plants of WT and *mil1-1* were transplanted for each replicate. At the fully mature stage, 5 plants of WT and *mil1-1* from each replicate were sampled. A total of 15 plants of WT and *mil1-1* were collected. Major agronomic traits, including plant height, number of tillers per plant, panicle length, panicle weight, branch number of panicles, number of filled grains per panicle, seed-setting rate, 1000-grain weight, and seed size, were measured.

### 4.2. Histochemical Assay

For DAB and NBT staining, leaves of WT and *mil1-1* sampled at the same positions during the tillering stage were soaked in 1 mg/mL 3,3′-diaminobenzidine (DAB) and 0.5 mg/mL of nitro-blue tetrazolium chloride (NBT) staining solution. The samples were stained for 12 h at room temperature, washed with water, and decolorized with fresh 95% ethanol. H_2_O_2_ and O^2−^ accumulation was observed under a microscope.

For trypan blue staining, leaves of WT and *mil1-1* in the same leaf position at the tillering stage were immersed in trypan blue staining solution. The samples were boiled in boiling water for 5 min and then left at room temperature in the dark for 12 h. The leaves were decolorized with 2.5 mg/mL chloral hydrate, blotted dry, and photographed under a stereomicroscope to observe the cell death phenomenon. Three biological replicates were performed during the histochemical assay.

### 4.3. Apoptosis Assay

To prepare the paraffin sections, the following procedure was used. Five to ten mm leaves from the spotted parts of *mil1-1* and the same parts of WT were sampled at the tillering stage. Then the samples were fixed in an FAA fixation solution containing formalin, acetic acid, and ethanol. After 24 h of fixation, the samples were sequentially immersed in 50%, 70%, 85%, 95%, 100%, and 100% ethanol for dehydration. Then, the samples were made transparent with xylene. After making the sample transparent, the samples were soaked in wax. After wax impregnation, the samples were embedded by packing the wax blocks for slicing. Then, the wax-wrapped samples were cut to the required thickness with a slicer. Next, the slices were placed on a slide for further study.

A TUNEL Apoptosis Assay Kit (One Step) (Beyotime, Shanghai, China, Cat. C1086) was used to detect DNA fragmentation. The paraffin sections were processed according to the instructions of the kit. In brief, the tissue sections were deparaffinized with xylene. After deparaffinization, the tissue sections were rehydrated in decreasing concentrations of ethanol (i.e., 100%, 95%, 85%, and 70%). After permeabilization with proteinase K, we washed the slides in a PBS buffer. PI and TUNEL detecting solutions were then added to the slides to label DNA and DNA fragments. Then, the signals of PI and TUNEL were observed under a confocal laser scanning microscope.

### 4.4. POD, CAT Activity Detection, and H_2_O_2_ Determination

Fresh leaves of WT and *mil1-1* at the tillering stage (DAS60) were selected for analysis using the protocol provided by the assay kit (Nanjing Jiancheng Institute of Biological Engineering, Nanjing, China). Peroxidase (POD) and catalase (CAT) activities and H_2_O_2_ content were measured according to the instructions of the kit (Cat. A007-1-1, A084-3-1, and A064-1-1). Three biological replicates were performed for each measurement. The protocols to examine POD and CAT enzyme activity and H_2_O_2_ concentration were as follows.

To analyze POD enzyme activity (Cat. A084-3-1), we weighed 0.2 g of leaves precisely, and a 1.8 mL physiological saline homogenous medium was added. The mixture was then homogenized in an ice water bath, and the homogenate was centrifuged at 3500 rpm for 10 min. We then extracted the supernatant for measurement by adding the proper amount of Reagent I to III provided by the kit. We mixed and incubated the solution in a water bath at 37 °C for 30 min. Then, Reagent IV was added. The mixture was then mixed thoroughly and centrifuged at 3500 rpm for 10 min. The supernatant was extracted, and the optical density (OD) value of each tube was measured at 420 nm with a 1 cm path length. The protein concentration of the samples was measured via the standard curve method. POD enzyme activity was calculated with the following formula:$$\begin{array}{c} \mathrm{POD}\;\mathrm{Enzyme}\;\mathrm{Activiy}\\ U/\mathrm{mg}\;\mathrm{prot} \end{array}=\frac{{\mathrm{OD}}_{\mathrm{Sample}}-{\;\mathrm{OD}}_{\mathrm{Reference}}}{\mathrm{MAC}\;\times \begin{array}{c} \mathrm{Path}\;\mathrm{Length}\\ 1\;\mathrm{cm} \end{array}}\times \frac{V_{\mathrm{total}}\left(4.0\mathrm{mL}\right)}{V_{\mathrm{sample}}\left(0.1\mathrm{mL}\right)}\;÷\;\begin{array}{c} t\\ 30\;\min \end{array}\times 1000\;÷\;\begin{array}{c} C_{\mathrm{Protein}}\\ \mathrm{mg}/\mathrm{mL} \end{array}$$
Note: MAC represents the milli-molar attenuation coefficient and is 12.0 mM^−1^·cm^−1^.One enzyme activity unit is defined as 1 μg substrate catalyzed per minute by enzymes within 1 mg tissue samples at 37 °C.

To analyze CAT enzyme activity (Cat. A007-1-1), we weighed 0.2 g of leaves precisely and added 1.8 mL physiological saline homogenous medium. We then homogenized the mixture in an ice water bath and centrifuged the homogenate at 2500 rpm for 10 min. We extracted the supernatant for measurement by adding the proper amount of Reagent 1 and 2 provided by the kit. The solution was mixed and incubated in a water bath at 37 °C for 1 min. Then, we added Reagents 3 and 4 and mixed the solution sufficiently. Next, the mixture was transferred into cuvettes with a 0.5 cm light path, and the OD values of all tubes were measured at 405 nm. CAT enzyme activity was calculated with the following formula:$$\begin{array}{c} \mathrm{CAT}\;\mathrm{Enzyme}\;\mathrm{Activiy}\\ U/\mathrm{mg}\;\mathrm{prot} \end{array}=\left({\mathrm{OD}}_{\mathrm{Reference}}-{\;\mathrm{OD}}_{\mathrm{Sample}}\right)\times 271\times \frac{1}{60\times {\;V}_{\mathrm{sample}}}\;÷\;\begin{array}{c} C_{\mathrm{Protein}}\\ \mathrm{mg}/\mathrm{mL} \end{array}$$
Note: 271 is reciprocal value of slope.1 µmol H_2_O_2_ decomposing per mg tissue protein per second is considered as 1 activity unit (U).

To analyze H_2_O_2_ concentration (Cat. A007-1-1), we weighed 0.2 g leaves precisely and added 1.8 mL physiological saline homogenous medium. The mixture was homogenized in an ice water bath, and the homogenate was centrifuged at 1000 rpm for 10 min. Then, the supernatant was extracted for measurement by adding the proper amount of pre-warmed Reagent I, H_2_O_2_ Standard Solution (163 mM H_2_O_2_), and Reagent II provided by the kit. The mixture was blended and transferred into cuvettes with a 1 cm light path. Then, the OD values of all tubes were measured at 405 nm. The H_2_O_2_ concentration was calculated with the following formula:$$\begin{array}{c} H_{2}O_{2}\;\mathrm{Content}\\ \mathrm{mmol}/g\;\mathrm{prot} \end{array}=\frac{{\mathrm{OD}}_{\mathrm{Sample}}-{\;\mathrm{OD}}_{\mathrm{Blank}}}{{\mathrm{OD}}_{\mathrm{Standard}}-{\;\mathrm{OD}}_{\mathrm{Blank}}}\times \begin{array}{c} C_{\mathrm{Standard}}\\ 163\;\mathrm{mM} \end{array}\;÷\;\begin{array}{c} \mathrm{Protein}\;\mathrm{Content}\\ g/L \end{array}$$

### 4.5. Bacterial Inoculation

At the tillering stage, leaves of WT and *mil1-1* were infected with *Xanthomonas oryzae* pv. *oryzae* (*Xoo*)to assess bacterial blight resistance. *Xoo* isolates of Zhe173, which are virulent to the WT, were used to assess the resistance of *mil1-1* plants to blight disease via the scissor-dipping method as described previously [40]. The strains were separately suspended in distilled water and adjusted to 109 viable cells/mL (OD600 = 1) [17]. Inoculation was performed by dipping scissors into the bacterial solution and cutting off about 2 cm of the leaf tips of the WT and *mil1-1*. Three distinct plants with at least three tillers from each plant were inoculated for each isolate. The disease incidence rate and lesion lengths were measured 2 weeks after inoculation. Three biological replicates were performed.

### 4.6. Gene Expression Analysis

To determine the expression of PR genes, the WT and *mil1-1* leaves were sampled at the seedling stage (14 days after sowing, DAS) before lesion formation and at the tillering stage (56 days after sowing) after lesion formation. Gene expression was examined using the method described previously [41,42]. In brief, total RNA was extracted from 100 mg of 14-day-old seedlings and adult rice plants using Takara RNAiso Plus (Takara Bio Inc., Otsu, Japan, Cat. No. 9108) for real-time RT-PCR. DNA contaminated in the isolated RNA was removed with RQ1 RNase-free DNase I (Promega, Madison, WI, USA, Cat. No. M6101). Three micrograms of RNA were used for first-strand cDNA synthesis with a RevertAid First Strand cDNA Synthesis Kit (Thermo Scientific, Waltham, MA, USA; Cat. No. K1621). Real-time RT-PCR was performed using Takara TB green premix Ex Taq (TB green premix Ex Taq, Takara Bio Inc., Cat. No. RR420A) and an Applied Biosystems QuantStudio 5 Real-Time PCR Instrument (Applied Biosystems, Foster City, CA, USA). For each sample, three technical replicates were created. Primers used in the assay are listed in Table 2. A total of three biological replicates were carried out. *OsACT1* (Os03g0718100) was employed as an endogenous control. The 2^−delta Ct^ formula was used to calculate the relative expression levels of genes. Primers used in the assay are listed in Table 2.

### 4.7. Genetic Analysis of mil1-1 Mutation and Map-Based Positional Cloning of MIL1

To perform the genetic analysis, the *mil1-1* mutant was crossed with the WT to produce F_1_; then, F_1_ individuals were self-pollinated to create F_2_. The F_2_ population was used for the segregation analysis of *mil1-1* mutations. The total number of plants and number of plants with *mil1-1* phenotypes were counted for the segregation ratio analysis and χ^2^ test.

To generate the mapping population of *mil1-1* (japonica rice background), the *mil1-1* mutant was crossed with the Japanese indica rice variety Habataki to produce F_1_; then, the F_1_ individuals were self-pollinated to produce F_2_. The segregated F_2_ population was utilized to positionally clone the *mil1-1* mutation. About 50 individual plants with the *mil1-1* mutant phenotype in the F_2_ population were used for primary mapping via BSA. Fine mapping was conducted by developing markers in the region of the *mil**1-1* mutation and enlarging the mapping population. After the mapping region was narrowed down, the PCR-sequencing strategy was used to find the mutation locus in *mil**1-1*.

### 4.8. Complementary Test of MIL1

For the complementary test, a 1464 bp *MIL1/OsHPL3* CDS fused to Green Fluorescence Protein (GFP) driven by the Cauliflower Mosaic Virus (CaMV) 35S constitutive promoter was cloned into the pCambia1300 binary vector. Then, the resulting *35S:MIL1c-GFP* plasmid was introduced into the *mil1-1* mutant via the agrobacterium (*Agrobacterium tumefaciens*)-mediated transformation of mature rice embryos as described previously [39,43]. In the T_2_ and T_3_ generations, we observed the phenotypes of complementary transgenic lines containing the *35S:MIL1c/OsHPL3c-GFP* construct.

### 4.9. Hormone and Stress Treatments

Two-week-old seedlings were treated with SA (100 µmol L^−1^), JA (100 µmol L^−1^), ABA (100 µmol L^−1^), NaCl (250 mmol L^−1^), and wound treatments for 6 h. Then, the treated seedlings were sampled to extract total RNA and conduct gene expression analysis, as described in 4.6. Three biological replicates were performed.

## Figures and Tables

**Figure 1 ijms-23-08853-f001:**
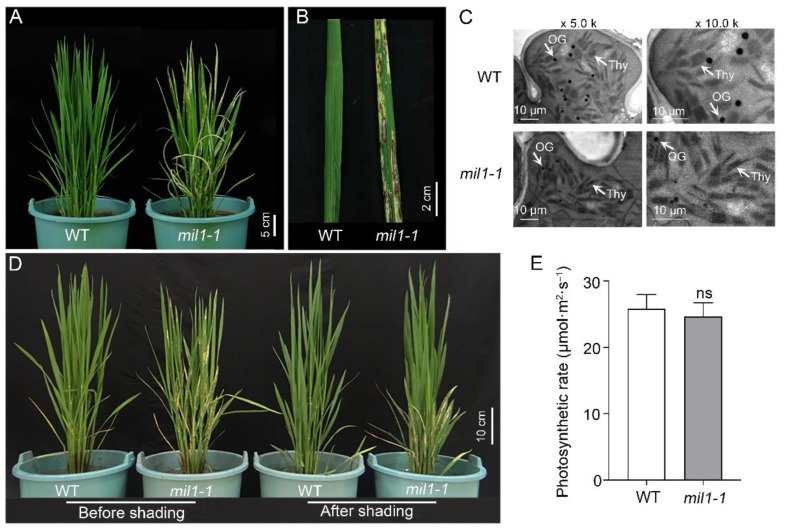
The *mil1-1* mutation confers lesion mimic leaf phenotype in rice. (**A**) Photos of WT and *mil1-1* plants at tillering stage, showing the spotted leaves of *mil1-1*. Scale bar is 20 cm. (**B**) Lesion mimic leaf of *mil1-1* compared to WT. Scale bar is 2 cm. (**C**) Transmission electron microscopy observation of chloroplasts in WT and *mil1-1* leaves. OG: osmiophilic plastoglobuli; Thy: Thylakoid lamellae. Scale bar is 10 μm. The left and right columns are different magnifications as indicated. (**D**) Effects of shading on lesion formation of WT and *mil1-1* leaves. Scale bar is 10 cm. (**E**) The photosynthetic rate of WT and *mil1-1*. Data in E are mean ± SD. A two-tailed unpaired *t*-test with Welch’s correction is used for statistical analysis. ns indicates no significant difference.

**Figure 2 ijms-23-08853-f002:**
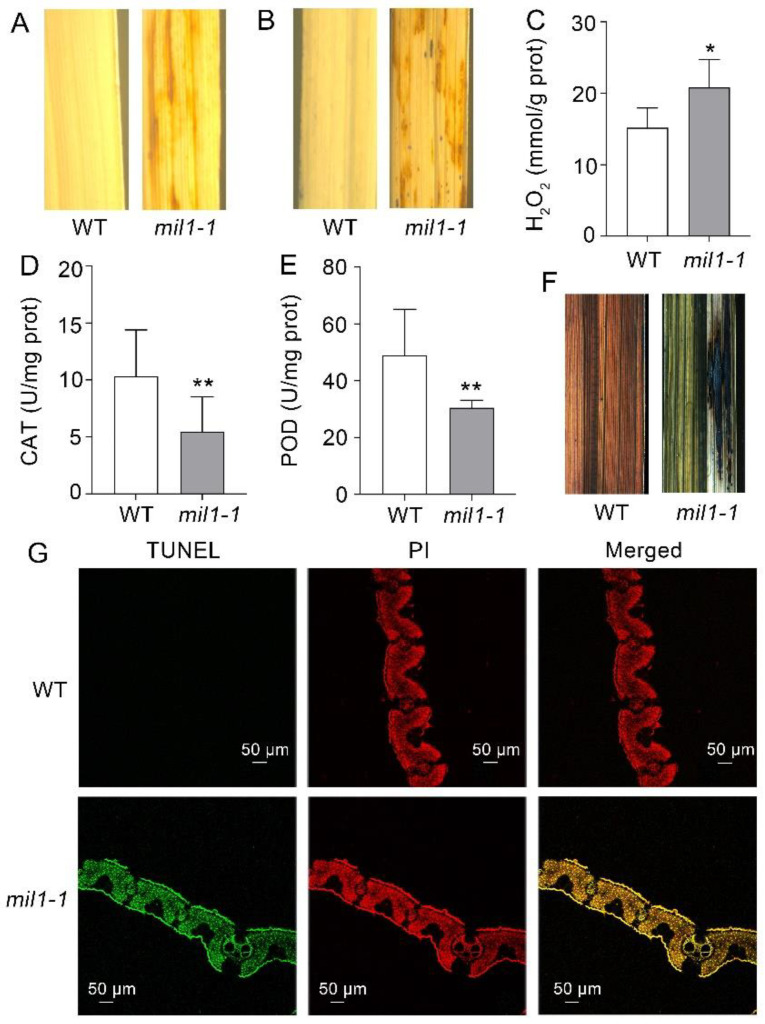
Histochemical analysis of ROS accumulation and cell death in WT and *mil1-1*. (**A**) DAB staining of leave in WT and *mil1-1*. (**B**) NBT staining of leave in WT and *mil1-1*. (**C**) H_2_O_2_ concentration in WT and *mil1-1*. (**D**,**E**) Enzyme activities of CAT (**D**) and POD (**E**) in WT and *mil1-1*. Data in (**C**–**E**) are means ± SD. prot: protein. U: unit. Three biological replicates were performed in the measurement (*n* = 3). A two-tailed unpaired t-test with Welch’s correction is used for statistical analysis. Asterisks indicate significant differences between treatments and respective controls at each time point (* *p* < 0.05, ** *p* < 0.01; Student’s *t* test). (**F**) Trypan blue staining of WT and *mil1-1* leaves. (**G**) DNA fragments detected by TUNEL assay. Scale bar is 50 μm. ROS: reactive oxygen species; DAB: diaminobenzidine; NBT: nitro-blue tetrazolium chloride; CAT: catalase; POD: peroxidase; TUNEL: terminal deoxynucleotidyl transferase-mediated dUTP nick-end labeling; PI: Propidium Iodide.

**Figure 3 ijms-23-08853-f003:**
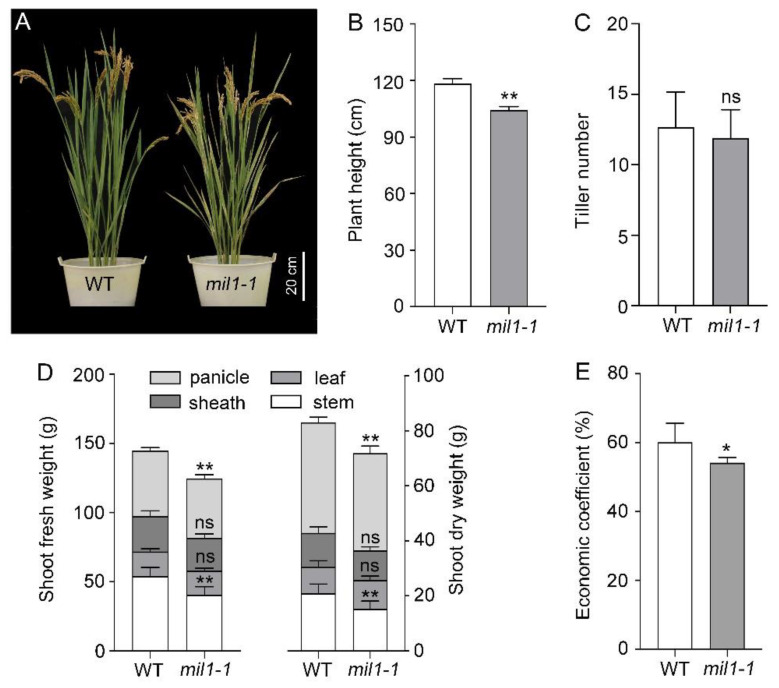
Biomass production in WT and *mil1-1*. (**A**) Morphological feature of WT and *mil1-1* at mature stage. (**B**–**E**) Agronomic traits, including plant height (**B**), tiller number (**C**), shoot fresh and dry weight (**D**), and economic coefficient (**E**), in WT and *mil1-1*. A two-tailed unpaired *t*-test with Welch’s correction is used for statistical analysis. Asterisks indicate significant differences between treatments and respective controls at each time point (* *p* < 0.05, ** *p* < 0.01; Student’s *t* test).

**Figure 4 ijms-23-08853-f004:**
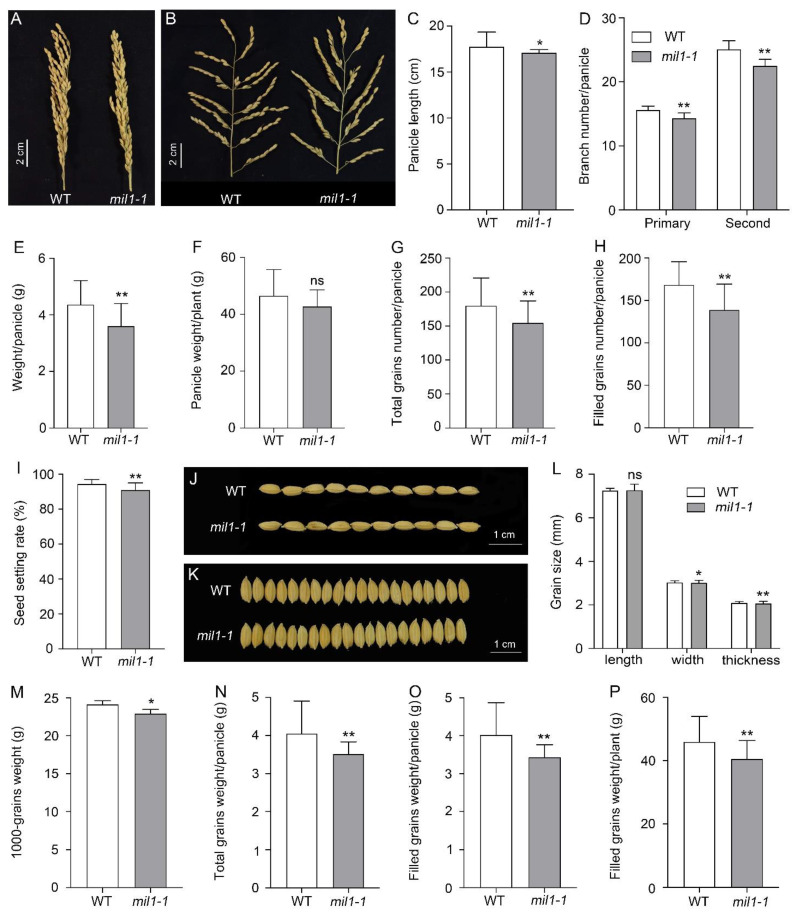
Agronomic traits related to yield formation in WT and *mil1-1*. (**A**–**F**) Panicle traits, including panicle length (**A**–**C**), branch number/panicle (**D**), weight/panicle (**E**), and panicle weight/plant (**F**) in WT and *mil1-1*; (**G**–**O**) Grain features, such as total grains number/panicle (**G**), filled grains number/panicle (**H**), seed setting rate (**I**), grain size (**J**–**L**), 1000-grains-weight (**M**), total grains weight/panicle (**N**), and filled grain weight/panicle (**O**). (**P**) Filled grain weight/plant. Data in (**C**–**I)** and (**L**–**P)** are means ± SD. A two-tailed unpaired *t*-test with Welch’s correction is used for statistical analysis. Asterisks indicate significant differences between treatments and respective controls at each time point (* *p* < 0.05, ** *p* < 0.01; Student’s *t* test).

**Figure 5 ijms-23-08853-f005:**
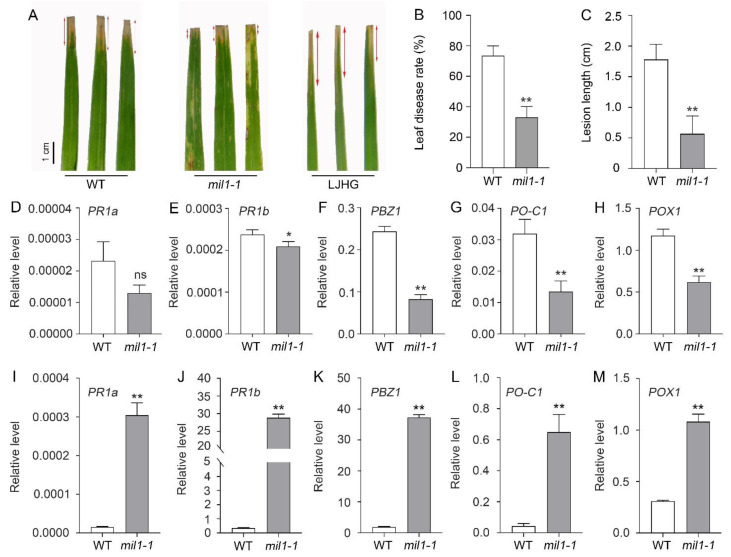
Mutation of *MIL1* enhances the resistance to bacterial blight in rice. (**A**) Phenotypes of WT and *mil1-1* inoculated with isolate of bacterial blight Zhe173. (**B**) Leaf disease rate of WT and *mil1-1* after inoculation. (**C**) Lesion length of WT and *mil1-1* after inoculation. (**D**–**H**) The expression of defense response genes detected by RT-qPCR before lesion formation, including *PR1a* (**D**), *PR1b* (**E**), *PBZ1/PR10* (**F**), *PO-C1* (**G**), and *POX1* (**H**). (**I**–**M**) The expression of defense response genes detected by RT-qPCR after lesion formation, including *PR1a* (**I**), *PR1b* (**J**), *PBZ1/PR10* (**K**), *PO-C1* (**L**), and *POX1* (**M**). *Rice ACTIN1* (*OsACT1*) was used as an internal control. Data in (**B**–**M)** are means ± SD. A two-tailed unpaired *t*-test with Welch’s correction is used for statistical analysis. Asterisks indicate significant differences between treatments and respective controls at each time point (* *p* < 0.05, ** *p* < 0.01; Student’s *t* test).

**Figure 6 ijms-23-08853-f006:**
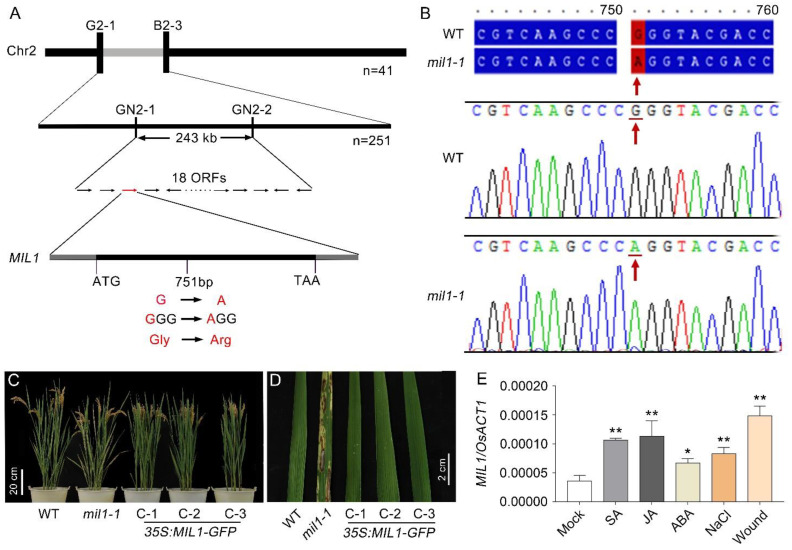
Positional cloning of *mil1-1* mutation and complementary test. (**A**) Map-based cloning of *MIL1.* Number under the horizontal chromosome bars transverse are recombinants and predicated 18 ORFs in the 243 kb region flanking by GN2-1 and GN2-2 markers. (**B**) Sequence alignment and the trace file of the mutation identified by sequencing in *mil1-1*. (**C**) Photos of WT, *mil1-1* and three complementary transgenic lines. Scale bar is 20 cm. (**D**) Leave comparison of WT, *mil1-1* and three complementary transgenic lines. Scale bar is 2 cm. (**E**) Expression of *MIL1*/*OsHPL3* detected by RT-qPCR in WT plants in different treatments. *Rice ACTIN 1* (*OsACT1*) was used as an internal control. Data in (**E**) are means ± SD. Asterisks indicate significant differences between treatments and respective controls at each time point (* *p* < 0.05, ** *p* < 0.01; Student’s *t* test).

**Table 1 ijms-23-08853-t001:** Genetic analysis of *mil1-1* mutation.

Cross Combination	Plants with WT Phenotype	Plants with *mil1-1* Phenotype	Total Plants	Plants with WT Phenotype: Plants with *mil1-1* Phenotype	χ^2^
*mil1-1* × WT	593	210	803	2.82:1 ≈ 3:1	0.5683
χ^2^ = 3.84, *p* = 0.05, df = 1		χ^2^ = 6.64, *p* = 0.01, df = 1		χ^2^ < χ^2^ (*p* = 0.05, df = 1)	

**Table 2 ijms-23-08853-t002:** Primers used in the study.

Name	Sequences	Note
OsACT1ReF	CTATGTTCCCTGGCATTGCT	OsACT1, RT-qPCR
OsACT1ReR	GGCGATAACAGCTCCTCTTG	
PR1AReF	AGGTTATCCTGCTGCTTGCT	PR1A, RT-qPCR
PR1AReR	AGTCCGAGTGCTCCAGCTT	
PR1BReF	AGAACTACGCCAGCCAGAGA	PR1B, RT-qPCR
PR1BReR	GTAGTGCCCGCACACCTT	
PBZ1ReF	ATGAAGCTTAACCCTGCCGC	PBZ1, RT-qPCR
PBZ1ReR	TCGAGCTCGTACTCCACCTT	
POC1ReF	GCTCTCTCAGGCGCGCACA	POC1, RT-qPCR
POC1ReR	TTCGACAGCAGGTTGGTGTA	
POX1ReF	GCTCTCTCAGGAGCACACAC	POX1, RT-qPCR
POX1ReR	CAGCAGGTTGCTGTAGTAGGC	
MIL1ReF	AAGGAGGAGGCCATCAACAA	MIL1, RT-qPCR
MIL1ReR	CATCCGGAGCACCTCGTAC	
MIL1cF	tctagaATGGTGCCGTCGTTCCCG *	*35S*:*MIL1*-*GFP* vector
MIL1cR	ggatccGCTGGGAGTGAGCTCCCGC	

* Letters in lower case indicate cutting sites used.

## Data Availability

All data generated or analyzed during this study are included in this published article.

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
