# Peer review of "A Single Amino Acid Substitution in MIL1 Leads to Activation of Programmed Cell Death and Defense Responses in Rice"

_ijms, 2022, doi:10.3390/ijms23168853_

Round 1

Reviewer 1 Report

The attached file contains my comments and suggestions.

Author Response

Please find the response to reviewer 1‘s comments from the file uploaded. Thanks!

Reviewer 2 Report

The manuscript of Yan and co-authors is devoted to the study of mil 1-1 mutants of rice, including obtaining the mutants themselves, comparing them with the wild type, identifying the basis of mutation and the response of plants to stress, including the level of illumination and bacterial infection. In my opinion, the authors have done an interesting complex work showing the mechanisms of plant response to infection and protection from reactive oxygen species. I have only two small clarifications.

1. Arginine is denoted as Arg = R. Explain, on line 228 - what does "Agd, G" mean?

2. Line 363, really 3 micrograms of RNA was used to build a k-DNA chain?

3. The question of determining the activity of enzymes. To determine gene expression, the authors selected several time points. Why did the authors take samples only once to determine the activity of enzymes? it seems to me that it would be more informative to look at the change in the activity of these enzymes over time, since a one-time measurement is rather uninformative. In addition, despite the fact that the authors provide a link to the instructions to the whales, it would be useful if the authors briefly gave a description of the methodology, how they determined activity, using which device, how the protein was determined, what were the units of activity expressed? gprot - what is it? per gram of protein? usually, the specific activity of enzymes is expressed in mg of protein. The conversion of activity to mg of protein gives too small a value. Accordingly, I would especially like to understand how the authors could measure it.

Author Response

Please find the response to reviewer 2 comments from the file uploaded. Please do not hesitate to contact us, if there is anything we should do!  Thanks!  

1

Round 2

Reviewer 1 Report

Comments are attached in the PDF file.

Author Response

Dear Reviewer,

Thank you for your comments on our revised manuscript. Please find the responses to the comments from the attached. If there is anything we should do please do not hesitate to contact us. Thank you for your time and consideration!

Regards,

Xiaoxue

Round 3

Reviewer 1 Report

The authors have incorporated the comments. I think the last paragraph of the introduction section needs some improvements.

This section of the introduction only presents the results in the current format, which is generally meant to be a place for hypothesis/question/goal of the study.

Author Response

Dear reviewer,

Thank you for you comments on the manuscript. Please find our response from the attached. If there is anything we should do, please let us know. Thank you for your time!

Regards,

Xiaoxue 
